# Operator Splitting Value Iteration

**Amin Rakhsha**[1,2] **Andrew Wang**[1,2] **Mohammad Ghavamzadeh**[3]

**Amir-massoud Farahmand**[2,1]

[1]Department of Computer Science, University of Toronto
[2]Vector Institute   [3]Google Research

## Abstract

We introduce new planning and reinforcement learning algorithms for discounted MDPs that utilize an approximate model of the environment to accelerate the convergence of the value function. Inspired by the splitting approach in numerical linear algebra, we introduce *Operator Splitting Value Iteration* (OS-VI) for both Policy Evaluation and Control problems. OS-VI achieves a much faster convergence rate when the model is accurate enough. We also introduce a sample-based version of the algorithm called OS-Dyna. Unlike the traditional Dyna architecture, OS-Dyna still converges to the correct value function in presence of model approximation error.

## 1 Introduction

Consider a planning problem for a discounted MDP with dynamics $\mathcal{P}$. Suppose that we have access to an approximate model $\hat{\mathcal{P}} \approx \mathcal{P}$ as well. For example, $\mathcal{P}$ might be a high-fidelity, but slow, simulator, and $\hat{\mathcal{P}}$ is a lower-fidelity, but fast, simulator. Or in the context of model-based reinforcement learning (MBRL), $\mathcal{P}$ is the unknown dynamics of a real-world system, from which we can only acquire expensive samples, and $\hat{\mathcal{P}}$ is a learned model, from which samples can be cheaply acquired. Can we use this approximate model $\hat{\mathcal{P}}$ to *accelerate* the computation of the value function of a policy $\pi$ (Policy Evaluation (PE) problem) or the optimal value function (Control problem)?

The Value Iteration (VI) algorithm and its approximate variant are fundamental algorithms in Dynamic Programming that can find the (approximate) value of a policy or the optimal value function. They are also the backbone of many reinforcement learning (RL) algorithms such as Temporal Difference Learning [Sutton, 1988], Fitted Value Iteration [Gordon, 1995, Ernst et al., 2005, Munos and Szepesvári, 2008], and Deep Q Network [Mnih et al., 2015]. Value Iteration, however, can be slow when the discount factor is close to 1, as its convergence rate is $O(\gamma^k)$. Moreover, even though we could use VI using $\hat{\mathcal{P}}$ instead of $\mathcal{P}$ to avoid expensive queries to $\mathcal{P}$, the obtained value function would converge to a solution different from the value function of the original MDP.

This paper proposes an algorithm called *Operator Splitting Value Iteration* (OS-VI) that benefits from an approximate model $\hat{\mathcal{P}}$ to potentially accelerate the convergence of the value function sequence to the value function with respect to (w.r.t.) the true model $\mathcal{P}$ (Section 3). This algorithm is for both PE (Section 3.1) and Control (Section 3.2) problems. The acceleration is not uniform though, and depends on how close $\hat{\mathcal{P}}$ is to $\mathcal{P}$ (Section 4).

A key inspiration behind OS-VI is the (matrix) splitting approach in the numerical linear algebra, which is used to iteratively solve large linear systems of equations [Varga, 2000, Saad, 2003, Golub and Van Loan, 2013]. With a proper choice of splitting, one may change the convergence rate of

36th Conference on Neural Information Processing Systems (NeurIPS 2022).

linear systems solvers. We show that the conventional VI for PE can be seen as a particular choice of splitting. This observation suggests that one may choose other forms of splitting as well in order to change the convergence rate. It turns out that we can choose a splitting that benefits from having access to $\hat{\mathcal{P}}$ (Section 2). The new splitting leads to OS-VI for PE. For the Control problem, the connection between solving linear system of equations and VI is not as straightforward anymore, as the former is linear, while the latter is not, but we can still get inspired from the splitting approach to design OS-VI for Control. The key step of such an algorithm is a new *policy improvement* step.

The form of the OS-VI algorithm opens up a connection to MBRL where the approximate model $\hat{\mathcal{P}}$ is learned using data. This leads to the OS-Dyna algorithm, inspired by a generic Dyna architecture [Sutton, 1990]. OS-Dyna is a hybrid of model-free and model-based algorithms. It uses the learned model in its inner planning loop, alike Dyna, but uses samples from the true model $\mathcal{P}$ in order to correct the effect of errors in the model. Existing MBRL algorithms would converge to an incorrect solution if the approximate model $\hat{\mathcal{P}}$ does not converge to the true model $\mathcal{P}$. This would be the case whenever model approximation error exists. On the other hand, OS-Dyna can still converge to the correct value function even when $\hat{\mathcal{P}}$ does not converge to $\mathcal{P}$. As far as we know, this is the first model-based RL algorithm with such property.

## 2 From value iteration to splitting-based linear system of equations solvers and back

We briefly describe the VI algorithm and the splitting methods for solving linear system of equations, and explain their connections. We consider a discounted Markov Decision Process (MDP) $(\mathcal{X}, \mathcal{A}, \mathcal{R}, \mathcal{P}, \gamma)$ [Bertsekas and Tsitsiklis, 1996, Szepesvári, 2010, Sutton and Barto, 2019]. We defer formal definitions to the supplementary material. We only mention that for a policy $\pi$, we denote by $\mathcal{P}^\pi$ its transition kernel, by $r^\pi : \mathcal{X} \to \mathbb{R}$ the expected value of its reward distribution, and by $V^\pi = V^\pi(\mathcal{R}, \mathcal{P})$ its state-value function. We also represent the optimal state-value function by $V^* = V^*(\mathcal{R}, \mathcal{P})$ and the optimal policy by $\pi^* = \pi^*(\mathcal{R}, \mathcal{P})$. The Bellman operator $T^\pi : \mathcal{B}(\mathcal{X}) \to \mathcal{B}(\mathcal{X})$ for policy $\pi$ and the Bellman optimality operator $T^* : \mathcal{B}(\mathcal{X}) \to \mathcal{B}(\mathcal{X})$ are[1]

$$(T^\pi V)(x) \triangleq r^\pi(x) + \gamma \int \mathcal{P}^\pi(\mathrm{d}y|x)V(y); \quad (T^*V)(x) \triangleq \max_{a \in \mathcal{A}} \left\{ r(x,a) + \gamma \int \mathcal{P}(\mathrm{d}y|x,a)V(y) \right\}.$$

These operators can be written more compactly as $T^\pi : V \mapsto r^\pi + \gamma \mathcal{P}^\pi V$ and $T^* : V \mapsto \max_\pi \{r^\pi + \gamma \mathcal{P}^\pi V\}$. The *greedy* policy at state $x \in \mathcal{X}$ is

$$\pi_g(x; V) \leftarrow \underset{a \in \mathcal{A}}{\mathrm{argmax}} \left\{ r(x,a) + \gamma \int \mathcal{P}(\mathrm{d}y|x,a)V(y) \right\}, \tag{2.1}$$

or more compactly, $\pi_g(V) \leftarrow \mathrm{argmax}_\pi T^\pi V$. We have $T^*V = T^{\pi_g(V)}V$, that is, the effect of the Bellman optimality operator $T^*$ applied to a value function $V$ is the same as applying the Bellman operator of the *greedy* policy w.r.t. $V$ to $V$.

### 2.1 Value Iteration

The value function $V^\pi$ and the optimal value function $V^*$ are the fixed points of the operators $T^\pi$ and $T^*$, respectively, and satisfy the Bellman equation. For the PE problem, this means that

$$V^\pi = r^\pi + \gamma \mathcal{P}^\pi V^\pi \Rightarrow (\mathbf{I} - \gamma \mathcal{P}^\pi)V^\pi = r^\pi. \tag{2.2}$$

There are several ways to compute the value function of a policy $\pi$ or the optimal value function $V^*$, including the iterative methods such as VI and Policy Iteration (PI) algorithms, or solving a linear system of equations (for PE) or linear programming (for Control). We focus on the VI algorithm in this work. VI repeatedly applies the Bellman operator to the most recent approximation of the value function: Given an initial value function $V_0$, it generates a sequence $(V_k)_{k \geq 0}$ as follows:

$$V_k \leftarrow \begin{cases} T^\pi V_{k-1}, & \text{(Policy Evaluation)} \\ T^*V_{k-1}. & \text{(Control)} \end{cases} \tag{2.3}$$

---

[1]For countable state and action spaces, the integrals are replaced by summations. We present OS-VI and its theoretical analysis for general state/action spaces, but limit our experiments to finite state/action problems.

VI for Control can be written in an equivalent form: At iteration $k$, we first compute the greedy policy $\pi_k \leftarrow \pi_g(V_{k-1})$ (policy improvement step), and then $V_k \leftarrow T^{\pi_k} V_{k-1}$. Therefore, the policy improvement step is obtained through finding a policy that is greedy w.r.t. the last value function $V_{k-1}$, that is, the best policy if we only look one step ahead. This form will be conductive for our later developments. As the Bellman operator is a $\gamma$-contraction w.r.t. the supremum norm, the convergence rate of $V_k$ to $V^\pi$ (or $V^*$) would be $O(\gamma^k)$. This rate can be slow when $\gamma$ is close to 1.

## 2.2 Matrix splitting for solving linear system of equations

The VI for PE can be seen as a (matrix) splitting-based iterative method to solve the linear system of equations (2.2). Consider the linear system $Az = b$, with $A \in \mathbb{R}^{d \times d}$ and $z, b \in \mathbb{R}^d$. Suppose that $A$ is decomposed to $A = M - N$ for some choices of $M, N \in \mathbb{R}^{d \times d}$ (more generally, $A$, $M$, and $N$ can be linear operators). Therefore, $z$ satisfies $Mz = Nz + b$. The splitting-based iterative approach defines the new approximation $z_k$ given the current $z_{k-1}$ by solving

$$Mz_k = Nz_{k-1} + b,$$

or equivalently,

$$z_k = M^{-1}(Nz_{k-1} + b). \tag{2.4}$$

To analyze the convergence of this iterative method, consider the error $e_k \triangleq z_k - z$. As $z = M^{-1}(Nz + b)$, we have $e_k = M^{-1}(Nz_{k-1} + b) - M^{-1}(Nz + b) = M^{-1}N(z_{k-1} - z)$, so the dynamics of the error is

$$e_k = M^{-1}Ne_{k-1} = (M^{-1}N)^2 e_{k-2} = \cdots = (M^{-1}N)^k e_0. \tag{2.5}$$

Let $G \triangleq M^{-1}N$. The norm of the sequence $(e_k)_{k \geq 1}$ can be upper bounded as

$$\|e_k\| = \|G^k e_0\| \leq \|G^k\| \|e_0\| \leq \|G\|^k \|e_0\|. \tag{2.6}$$

The errors are (norm-) convergent if $\|G\| = \|M^{-1}N\| < 1$, for some choice of norm. More generally, the necessary and sufficient condition for convergence is that the spectral radius $\rho(G)$, the maximum absolute value of eigenvalues of $G$, is smaller than one, see e.g., Theorem 4.1 of Saad [2003] or Theorem 11.2.1 of Golub and Van Loan [2013].[2] The convergence is faster if the spectral radius (or norm) is closer to zero.

The success of this iterative procedure depends on how we choose $M$ and $N$ such that the norm (or spectral radius) is as small as possible. Also we want to choose an $M$ such that solving $Mz_k = Nz_{k-1} + b$ is not very expensive. For example, if $M$ is an identity matrix $\mathbf{I}$, we get that $N = \mathbf{I} - A$, and the iteration becomes $z_k = (\mathbf{I} - A)z_{k-1} + b$. This iteration is convergent if $\rho(\mathbf{I} - A) < 1$. Other commonly used choices lead to the Jacobi and Gauss-Seidel methods that are described in the supplementary material.

We are now ready to make the connection between splitting-based iterative methods and VI for PE. If we choose $A = \mathbf{I} - \gamma \mathcal{P}^\pi$, we see that equation $AV^\pi = r^\pi$ is indeed the Bellman equation for policy $\pi$ (2.2). The VI for PE, which is $V_k = \gamma \mathcal{P}^\pi V_{k-1} + r^\pi = (\mathbf{I} - A)V_{k-1} + r^\pi$, corresponds to the choice of $M = \mathbf{I}$ and $N = \gamma \mathcal{P}^\pi$. This brings up the question of whether it is possible to split $A$ differently so that the resulting VI-like procedure has better convergence properties. We next suggest a particular choice.

## 3 Operator splitting value iteration algorithm

We introduce the Operator Splitting Value Iteration (OS-VI) algorithm. We start from the PE problem and develop the Control version based on that. We also present a visualization of how OS-VI works.

### 3.1 OS-VI for policy evaluation

Given a policy $\pi$, true model $\mathcal{P}$, and approximate model $\hat{\mathcal{P}}$, we split $\mathbf{I} - \gamma \mathcal{P}^\pi$ to $M^\pi$ and $N^\pi$ as

$$M^\pi = \mathbf{I} - \gamma \hat{\mathcal{P}}^\pi \qquad , \qquad N^\pi = \gamma(\mathcal{P}^\pi - \hat{\mathcal{P}}^\pi).$$

---

[2] For any matrix norm, we have $\rho(G) \leq \|G\|$, so the condition on the norm is sufficient, but not necessary. Our analysis will be norm-based.

Following the recipe of (2.4), the OS-VI algorithm for PE would be

$$V_k \leftarrow (\mathbf{I} - \gamma \hat{\mathcal{P}}^\pi)^{-1} \Big[ r^\pi + \gamma(\mathcal{P}^\pi - \hat{\mathcal{P}}^\pi) V_{k-1} \Big], \tag{3.1}$$

starting from an initial $V_0$.[3] To gain more intuition and prepare for further developments, we define a few notations. We define the *Varga operator* $S^\pi : \mathcal{B}(\mathcal{X}) \to \mathcal{B}(\mathcal{X})$, named after Richard S Varga (1928 – 2022) who has made significant contributions to matrix analysis, as the mapping between the space of all bounded functions over $\mathcal{X}$ to the same space as

$$S^\pi : V \mapsto (\mathbf{I} - \gamma \hat{\mathcal{P}}^\pi)^{-1} \Big[ r^\pi + \gamma(\mathcal{P}^\pi - \hat{\mathcal{P}}^\pi) V \Big].$$

Observe that (3.1) can be compactly written as

$$V_k \leftarrow S^\pi V_{k-1}. \tag{3.2}$$

It is not difficult to see that $S^\pi V^\pi = V^\pi$, i.e., the value function of a policy $\pi$ is a fixed-point of the Varga operator $S^\pi$. This and other properties of the Varga operator are shown in the supplementary material.

Given any value function $V$, define an auxiliary reward function $\bar{r}_V : \mathcal{X} \times \mathcal{A} \to \mathbb{R}$ as

$$\bar{r}_V(x, a) \triangleq r(x, a) + \gamma \int \Big( \mathcal{P}(\mathrm{d}y|x, a) - \hat{\mathcal{P}}(\mathrm{d}y|x, a) \Big) V(y). \tag{3.3}$$

Similar to $r^\pi$, we define the notation $\bar{r}_V^\pi : \mathcal{X} \to \mathbb{R}$ as $\bar{r}_V^\pi(x) = \bar{r}_V(x, \pi(x))$ for a deterministic policy $\pi$ (and similarly for a stochastic policy). With this notation, the effect of applying $S^\pi$ to $V$ is

$$S^\pi V = (\mathbf{I} - \gamma \hat{\mathcal{P}}^\pi)^{-1} \bar{r}_V^\pi.$$

This is the value function of following $\pi$ in an MDP with dynamics $\hat{\mathcal{P}}$ and reward $\bar{r}_V$. Therefore, at each iteration of OS-VI (PE), we evaluate the policy $\pi$ according to the approximate dynamics, and a reward function that consists of the original reward $r$ and the correction term $\gamma(\mathcal{P} - \hat{\mathcal{P}}) V_{k-1}$. The computation of this value function is a standard PE problem with the approximate model. For instance, we may use another VI (PE) with dynamics $\hat{\mathcal{P}}$ to solve it: We initialize $U_0 \leftarrow V$, and then for $i \geq 1$, we set $U_i \leftarrow \bar{r}_V^\pi + \gamma \hat{\mathcal{P}}^\pi U_{i-1}$. This converges to $S^\pi V = (\mathbf{I} - \gamma \hat{\mathcal{P}}^\pi)^{-1} \bar{r}_V^\pi$ with the usual rate of $O(\gamma^i)$. Note that aside from the computation of $\bar{r}_V^\pi$, which requires querying $\mathcal{P}$ in order to compute the $\mathcal{P}^\pi V$ term, this iterative process only uses the approximate model $\hat{\mathcal{P}}$, which is assumed to be cheap to access.

What is the benefit of this OS-VI procedure? If the approximate model $\hat{\mathcal{P}}$ is close to the true dynamics $\mathcal{P}$, this leads to a faster convergence of $V_k$ to $V^\pi$, as shall be quantified soon. The faster convergence is in terms of the number of queries to $\mathcal{P}$, which is assumed to be expensive. To see this, consider the hypothetical case that $\hat{\mathcal{P}}$ is exactly the same as $\mathcal{P}$, for example, if the cheap simulator happens to perfectly match the reality. Then, $S^\pi V = (\mathbf{I} - \gamma \mathcal{P}^\pi)^{-1}(r^\pi + 0V) = V^\pi$, and the value function for the original MDP is obtained in one iteration of OS-VI. Of course, we often can only hope for $\hat{\mathcal{P}} \approx \mathcal{P}$. In Section 4, we study the impact of error in $\hat{\mathcal{P}}$ on the convergence rate of OS-VI in more details, and show that the convergence of OS-VI can be much faster than classic algorithms even if $\hat{\mathcal{P}}$ is only a close approximation of $\mathcal{P}$.

## 3.2 OS-VI for control

The VI for Control can be seen as an iterative procedure that computes the greedy policy $\pi_k \leftarrow \pi_g(V_{k-1}) = \arg\max_\pi T^\pi V_{k-1}$ in its policy improvement step, and then uses one step of the Bellman operator w.r.t. the obtained policy $\pi_k$ to compute the new estimate of the value function $V_k \leftarrow T^{\pi_k} V_{k-1}$, as described after (2.3). The OS-VI algorithm for Control follows a similar structure with the difference that **1)** the improved policy is the optimizer of the Varga operator, and not the Bellman

---

[3]Although splitting is originally studied mostly in the context of linear algebra and matrices, we are applying the idea more generally. We are not assuming that the state space $\mathcal{X}$ is finite, and we allow it to be more general, such as a subset of $\mathbb{R}^d$. Consequently, $M^\pi$, $N^\pi$, $\mathcal{P}^\pi$, etc. are operators rather than matrices.

operator, and **2)** the new value function is obtained by applying the Varga operator of the newly obtained policy. To be concrete, given a value function $V$, define the *S-improved policy*

$$\pi_V(V) \triangleq \underset{\pi}{\arg\max}\, S^\pi V [= (\mathbf{I} - \gamma \hat{\mathcal{P}}^\pi)^{-1} \bar{r}_V^\pi]. \tag{3.4}$$

This policy is the *optimal* policy for the auxiliary MDP $(\mathcal{X}, \mathcal{A}, \bar{r}_V, \hat{\mathcal{P}}, \gamma)$. We also define the *Varga optimality operator* $S^* : \mathcal{B}(\mathcal{X}) \to \mathcal{B}(\mathcal{X})$ as

$$S^* : V \mapsto \max_\pi S^\pi V.$$

The function $S^*V$ is equal to $S^{\pi_V(V)}V$, i.e., the Varga operator of the $S$-improved policy w.r.t. $V$ applied to a value function $V$ (compare it with $T^*V = T^{\pi_g(V)}V$).

The OS-VI (Control) is then simply

$$V_k \leftarrow S^* V_{k-1}, \tag{3.5}$$

which in its expanded form, consists of the following two steps:

$$\pi_k \leftarrow \pi_V(V_{k-1}), \qquad \text{(policy improvement)}. \tag{3.6}$$

$$V_k \leftarrow S^{\pi_k} V_{k-1} \left[= (\mathbf{I} - \hat{\mathcal{P}}^{\pi_k})^{-1}(r^{\pi_k} + \gamma(\mathcal{P}^{\pi_k} - \hat{\mathcal{P}}^{\pi_k})V_{k-1})\right], \text{(partial policy evaluation)}. \tag{3.7}$$

Comparing the $S$-improved policy (3.4) used in the policy improvement step (3.6) of OS-VI with the conventional greedy policy (2.1) is insightful. The greedy policy is $\arg\max_\pi T^\pi V$. Expanding $T^\pi V$, we see that the greedy policy is the maximizer of $r^\pi + \gamma \mathcal{P}^\pi V$. The function $r^\pi + \gamma \mathcal{P}^\pi V$ is a single-step bootstrapped estimate of the value of $V^\pi$, and its maximizer, the greedy policy, is in general different from the optimal policy, which maximizes $V^\pi$. On the other hand, the $S$-improved policy $\pi_V(V)$ solves a full MDP with an approximate model $\hat{\mathcal{P}}$ and a reward function that has both the original reward $r$ and the correction term $\gamma(\mathcal{P} - \hat{\mathcal{P}})V$. In the special case that $\hat{\mathcal{P}} = \mathcal{P}$, the correction term is zero, and $\pi_V(V)$ would be the optimal policy $\pi^*$ for the original MDP. As often $\hat{\mathcal{P}} \approx \mathcal{P}$, the value function of policy $\pi_V(V)$ is not exactly the optimal value. The partial policy evaluation step (3.7) updates the value function to a value that is closer to the optimal value function.

**Remark.** The use of matrix splitting-based ideas, either explicitly or implicitly, in the context of dynamic programming is not completely novel to this work. Kushner and Kleinman [1971] is one of the earliest paper that mentions the Jacobi and Gauss-Seidel procedures for computing the value function. Porteus [1975] proposes several transformations to the reward and probability transition matrix with the goal of improving the computational cost of solving the transformed MDP. One of the transformations, called *pre-inverse transform*, has some similarities with the operator splitting of this work. The end result, however, is different. Bacon and Precup [2016] offer a matrix splitting perspective on planning with options. The connection between multi-step models and matrix splitting is developed in Chapter 4 of Bacon [2018]. Refer to the supplementary material for more discussion.

### 3.3 Visualizing how OS-VI works

To present some intuition on how OS-VI works, we visualize the value function trajectories of several algorithms, including OS-VI, on a 2-state MDP, in Figure 1. We consider the policy evaluation for the dynamics $\mathcal{P}^\pi = \left[\begin{smallmatrix} 0.9 & 0.1 \\ 0.1 & 0.9 \end{smallmatrix}\right]$ with the reward $r^\pi = \left(\begin{smallmatrix} 1 \\ -0.5 \end{smallmatrix}\right)$ and $\gamma = 0.9$. We consider two approximate models: a relatively accurate $\hat{\mathcal{P}}^\pi_{\text{accurate}} = \left[\begin{smallmatrix} 0.85 & 0.15 \\ 0.05 & 0.95 \end{smallmatrix}\right]$, and an inaccurate $\hat{\mathcal{P}}^\pi_{\text{inaccurate}} = \left[\begin{smallmatrix} 0.6 & 0.4 \\ 0.3 & 0.7 \end{smallmatrix}\right]$.

In addition to OS-VI (PE), the first algorithm is the conventional VI (PE), which repeatedly applies the Bellman operator according to the true model $\mathcal{P}^\pi$ to the most recent approximation of the value function. We use $T^\pi_{\mathcal{P}}$ to refer to its Bellman operator and to label the corresponding trajectory in the value space. This algorithm converges to the true value function $V^\pi$. The second algorithm is VI (PE) that uses $\hat{\mathcal{P}}^\pi$ as the model. This procedure is the basis of the Dyna architecture. We use $T^\pi_{\hat{\mathcal{P}}}$ to refer to its Bellman operator and to label the corresponding trajectory in the value space. Due to the error of $T^\pi_{\hat{\mathcal{P}}}$ compared to $T^\pi_{\mathcal{P}}$, the algorithm converges to a wrong value function $\hat{V}^\pi$, as both figures show. We observe that even when the model is relatively accurate as in Figure 1a, the converged value function is quite wrong. This illustrates one limitation of the conventional model-based RL algorithms where an inaccurate model may lead to significantly inaccurate estimate of the value function.

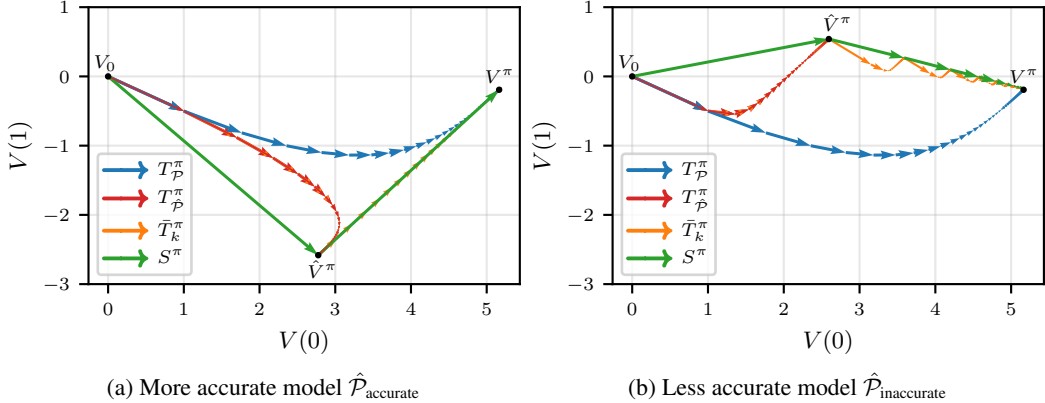

(a) More accurate model $\hat{\mathcal{P}}_{\text{accurate}}$  (b) Less accurate model $\hat{\mathcal{P}}_{\text{inaccurate}}$

Figure 1: The value function trajectories of VI (PE) with the true model ($T_{\mathcal{P}}^{\pi}$), VI (PE) with approximate model $T_{\hat{\mathcal{P}}}^{\pi}$, and OS-VI (PE) ($S^{\pi}$; and $\bar{T}_k^{\pi}$ for its inner loop) for a 2-state problem.

The OS-VI algorithm repeatedly applies the Varga operator $S^{\pi}$ to the most recent approximation of the value function. As discussed earlier, each computation of $S^{\pi}V_{k-1}$ corresponds to solving an auxiliary MDP $(\mathcal{X}, \mathcal{A}, \bar{r}_{V_{k-1}}, \hat{\mathcal{P}}, \gamma)$. We denote the Bellman operator of this auxiliary MDP by $\bar{T}_k^{\pi}$. The figures show the trajectory generated by the iterative application of $S^{\pi}$ on the most recent value function as well as the trajectory for solving each auxiliary MDP, indicated by $\bar{T}_k^{\pi}$. We observe that the OS-VI algorithm converges to the correct value function despite using an incorrect model. When the model is more accurate, very few iterations of OS-VI gets a value close to $V^{\pi}$ (two iterations in Figure 1a); when the model is less accurate, a few more iterations are needed. Compared to VI, at least in these examples, the total number of iterations of OS-VI is significantly smaller.

When the initial value function is $V_0 = 0$, the result of the first iteration of OS-VI is the same value function computed by the VI with the wrong model $\hat{\mathcal{P}}^{\pi}$. This is because $V_1 \leftarrow S^{\pi}V_0 = (\mathbf{I} - \gamma\hat{\mathcal{P}}^{\pi})^{-1}\bar{r}_{V_0}^{\pi}$ and $\bar{r}_{V_0}^{\pi} = r^{\pi}$, so $V_1 = (\mathbf{I} - \gamma\hat{\mathcal{P}}^{\pi})^{-1}r^{\pi}$, the same solution as the value function obtained using the approximate model $\hat{\mathcal{P}}$. In these figures, this shows itself by the overlapping of the red arrows followed by $T_{\hat{\mathcal{P}}}^{\pi}$ and the first segment of the orange arrows, which are generated by the repeated application of $\bar{T}_1^{\pi}$. In further iterations of OS-VI, the auxiliary MDPs change and the path followed by $\bar{T}_k^{\pi}$ ($k \geq 2$) deviates from the solution of the VI with the wrong model.

## 4 Convergence analysis of operator splitting value iteration

In this section, we present the convergence analysis of OS-VI. Our results show that OS-VI has an $O(\gamma'^k)$ convergence rate for an effective discount factor $\gamma'$ that depends on the error between $\hat{\mathcal{P}}$ and $\mathcal{P}$. For small enough error, $\gamma' < \gamma$ and OS-VI has a faster convergence rate compared to the classic VI, Policy Iteration (PI), and Modified Policy Iteration (MPI), which all have $O(\gamma^k)$ behaviour. We provide results for both the $L_{\infty}$ and $L_p$ norms.

### 4.1 Convergence of OS-VI for policy evaluation

We study the convergence behaviour of OS-VI (PE) in presence of error in value updates. Specifically, we consider that at each iteration $k$, the update (3.2) has an error, i.e.,

$$V_k = S^{\pi}V_{k-1} + \epsilon_k^{\text{value}} \tag{4.1}$$

The error $\epsilon_k^{\text{value}}$ encompasses various sources of errors that might occur in solving the auxiliary MDP $(\mathcal{X}, \mathcal{A}, \bar{r}_{V_{k-1}}, \hat{\mathcal{P}}, \gamma)$. One source is the function approximation error due to using a function approximator to represent $V_k$, which is often required in large state spaces. Another is the estimation (i.e., statistical) error caused due to having a finite number of samples, instead of direct access to $\mathcal{P}$, in the RL setting. Refer to Munos and Szepesvári [2008], Antos et al. [2008], Farahmand et al. [2016], Chen and Jiang [2019], Fan et al. [2019] for the discussion of function approximation and estimation errors in the RL context. In this work, we do not analyze how the number of samples, the function

approximator, etc. affect the errors $\epsilon_k^{\text{value}}$. We offer error propagation results similar to Munos [2007] (approximate VI), Munos [2003] (approximate PI), and Scherrer et al. [2015] (approximate modified PI) for approximate OS-VI.

To study the convergence behaviour of OS-VI (PE), let $G^\pi = (\mathbf{I} - \gamma\hat{\mathcal{P}}^\pi)^{-1}\gamma(\mathcal{P}^\pi - \hat{\mathcal{P}}^\pi)$. We use the fact that $S^\pi V^\pi = V^\pi$ and write

$$\|V^\pi - V_k\|_\infty = \left\|S^\pi V^\pi - S^\pi V_{k-1} - \epsilon_k^{\text{value}}\right\|_\infty = \left\|G^\pi(V^\pi - V_{k-1}) - \epsilon_k^{\text{value}}\right\|_\infty$$
$$\leq \|G^\pi\|_\infty \|V^\pi - V_{k-1}\|_\infty + \left\|\epsilon_k^{\text{value}}\right\|_\infty. \tag{4.2}$$

Now, we have that

$$\|G^\pi\|_\infty = \left\|(\mathbf{I} - \gamma\hat{\mathcal{P}}^\pi)^{-1}\gamma(\mathcal{P}^\pi - \hat{\mathcal{P}}^\pi)\right\|_\infty \leq \frac{\gamma}{1-\gamma}\left\|\mathcal{P}^\pi - \hat{\mathcal{P}}^\pi\right\|_\infty, \tag{4.3}$$

where we used the fact that for any square matrix $F$ with a matrix norm $\|F\|_p < 1$, it holds that $\|(\mathbf{I} - F)^{-1}\|_p \leq \frac{1}{1-\|F\|_p}$ (see Lemma 2.3.3 of Golub and Van Loan 2013), and that the supremum norm of a stochastic matrix $\hat{\mathcal{P}}^\pi$ is 1. Assuming that $\|\epsilon_k^{\text{value}}\|_\infty \leq \epsilon^{\text{value}}$ for all $k \geq 1$ and defining effective discount factor $\gamma' = \frac{\gamma}{1-\gamma}\|\mathcal{P}^\pi - \hat{\mathcal{P}}^\pi\|_\infty$, the upper bounds (4.2) and (4.3) lead to $\|V^\pi - V_k\|_\infty \leq \gamma'^k \|V^\pi - V_0\|_\infty + \frac{1-\gamma'^k}{1-\gamma'}\epsilon^{\text{value}}$.

A few remarks are in order. First, whenever $\gamma' < \gamma$, this is guaranteed to be faster than the convergence rate of the conventional VI, which is $O(\gamma^k)$. This happens if $\|\mathcal{P}^\pi - \hat{\mathcal{P}}^\pi\|_\infty < 1 - \gamma$. If the model is very accurate, we obtain much faster rate than VI's. Since each iteration $k$ corresponds to a query to the true model $\mathcal{P}$, a faster rate entails that the algorithm requires fewer number of queries to the expensive model to reach the same level of accuracy.

Second, although the model error $\|\mathcal{P}^\pi - \hat{\mathcal{P}}^\pi\|_\infty$ is a reasonable choice to measure the distances between distributions (it is the maximum of the Total Variation distance between $\mathcal{P}^\pi(\cdot|x)$ and $\hat{\mathcal{P}}^\pi(\cdot|x)$ over $x$, which itself can be upper bounded by their KL divergence; see the supplementary material), it is somewhat conservative as it takes the supremum over the state space. Likewise, requiring $\|\epsilon_k^{\text{value}}\|_\infty$ to be small is also conservative, as approximating $S^\pi V_{k-1}$ using a function approximator given samples (RL setting) often leads to an $L_p$-norm type of guarantee. We now provide a different analysis to address these issues.

To present the $L_p$-norm result, we need to define some notations. First, we define the conditional discounted future-state distribution of policy $\pi$ under $\hat{\mathcal{P}}$ as the following probability distribution: Given a measurable set $B$, we have $\hat{\eta}^\pi(B|x) = (1-\gamma)\sum_{t=0}^\infty \gamma^t \mathbb{P}\left(X_t \in B|X_0 = x, \pi, \hat{\mathcal{P}}\right)$, where the chain $(X_t)_{t\geq 0}$ starts from state $x$ and evolves by following policy $\pi$ under transitions $\hat{\mathcal{P}}$. For an arbitrary distribution $\rho$ over the state space, we define the *discounted future-state distribution concentration coefficient of the approximate model* as

$$\hat{C}^\pi(\rho)^2 = \frac{1}{\gamma^2} \int \rho(\mathrm{d}x) \left\|\frac{\mathrm{d}\hat{\eta}^\pi(\cdot|x)}{\mathrm{d}\rho}\right\|_\infty^3. \tag{4.4}$$

Here $\frac{\mathrm{d}\hat{\eta}^\pi(\cdot|x)}{\mathrm{d}\rho}$ is the Radon–Nikodym derivative of the distribution $\hat{\eta}^\pi(\cdot|x)$ w.r.t. the distribution $\rho$. It is assumed that for any $x \in \mathcal{X}$, $\hat{\eta}^\pi(\cdot|x) \ll \rho$, i.e., $\hat{\eta}^\pi(\cdot|x)$ is absolutely continuous w.r.t. $\rho$ (otherwise, the coefficient would be set to infinity). This coefficient measures how concentrated the distribution $\hat{\eta}^\pi(\cdot|x)$ is compared to $\rho$. This is weighted according to the state distribution $\rho$. Similar concentrability coefficients, but not exactly this one, have appeared in the error propagation results [Kakade and Langford, 2002, Munos, 2003, 2007, Farahmand et al., 2010, Scherrer et al., 2015]. Finally, we define the weighted $\chi^2$-divergence of $\hat{\mathcal{P}}^\pi$ and $\mathcal{P}^\pi$ as

$$\chi^2_\rho(\mathcal{P}^\pi \,\|\, \hat{\mathcal{P}}^\pi) \triangleq \int \rho(\mathrm{d}x)\chi^2\left(\mathcal{P}^\pi(\cdot|x) \,\|\, \hat{\mathcal{P}}^\pi(\cdot|x)\right) = \int \rho(\mathrm{d}x) \int \frac{\left|\hat{\mathcal{P}}^\pi(\mathrm{d}y|x) - \mathcal{P}^\pi(\mathrm{d}y|x)\right|^2}{\hat{\mathcal{P}}^\pi(\mathrm{d}y|x)}.$$

This notion of model error is less strict in requiring accurate approximation $\mathcal{P}$ in all states. Usually only a subset of the state space is important or even reachable in a problem. The above model error can focus on only specific areas of the state space through the choice of distribution $\rho$.

We are now ready to present the main theorem for the approximate OS-VI (PE).

**Theorem 1.** *Consider the approximate OS-VI algorithm for PE (4.1). Let $\|\cdot\|_\star$ be either the supremum norm $\|\cdot\|_\infty$ ($\star = \infty$) or $\|\cdot\|_{4,\rho}$ for $\rho$ being an arbitrary distribution over the state space ($\star = 4, \rho$). Assume that $\|\epsilon_k^{value}\|_\star \leq \epsilon^{value}$ for all $k \geq 1$. Furthermore, define the effective discount factor as*

$$\gamma' = \frac{\gamma}{1-\gamma} \begin{cases} \left\|\mathcal{P}^\pi - \hat{\mathcal{P}}^\pi\right\|_\infty & (\star = \infty), \\ \sqrt{\hat{C}^\pi(\rho)\chi_\rho^2(\mathcal{P}^\pi \| \hat{\mathcal{P}}^\pi)} & (\star = 4, \rho). \end{cases}$$

*For any $k \geq 0$, we have $\|V^\pi - V_k\|_\star \leq \gamma'^k \|V^\pi - V_0\|_\star + \frac{1-\gamma'^k}{1-\gamma'} \cdot \epsilon^{value}$.*

### 4.2 Convergence of OS-VI for control

We turn to analyzing OS-VI for Control. We consider two types of errors: The first is the error between the computed value function and the true optimal value function of the auxiliary MDP, i.e., $V_k - S^* V_{k-1}$. The second is the suboptimality of obtained policy compared to the optimal policy of the auxiliary MDP, i.e., $S^{\pi_k} V_{k-1} - S^* V_{k-1}$. Concretely, we have

$$V_k = S^* V_{k-1} + \epsilon_k^{value}, \tag{4.5}$$

$$S^{\pi_k} V_{k-1} = S^* V_{k-1} + \epsilon_k^{policy}. \tag{4.6}$$

We have the following result for the approximate OS-VI (Control).

**Theorem 2.** *Consider the approximate OS-VI algorithm for control (4.5)-(4.6). Let $\|\cdot\|_\star$ be either the supremum norm $\|\cdot\|_\infty$ ($\star = \infty$) or $\|\cdot\|_{4,\rho}$ for $\rho$ being an arbitrary distribution over the state space ($\star = 4, \rho$). For any $k \geq 1$, let $\Pi_k = \{\pi^*, \pi_k\} \cup \{\pi_V(V_{i-1}) : 1 \leq i < k\}$. Assume that $\|\epsilon_k^{value}\|_\star \leq \epsilon^{value}$ for all $k \geq 1$. Furthermore, define the effective discount factor as*

$$\gamma' = \frac{\gamma}{1-\gamma} \begin{cases} \max_{\pi \in \Pi_k} \left\|\mathcal{P}^\pi - \hat{\mathcal{P}}^\pi\right\|_\infty & (\star = \infty), \\ \max_{\pi \in \Pi_k} \sqrt{\sqrt{2}\,\hat{C}^\pi(\rho)\chi_\rho^2(\mathcal{P}^\pi \| \hat{\mathcal{P}}^\pi)} & (\star = 4, \rho). \end{cases}$$

*We then have*

$$\|V^{\pi_k} - V^*\|_\star \leq \frac{2\gamma'^k}{1-\gamma'}\|V_0 - V^*\|_\star + \frac{2\gamma'(1-\gamma'^{k-1})}{(1-\gamma')^2}\epsilon^{value} + \frac{1}{1-\gamma'}\left\|\epsilon_k^{policy}\right\|_\star.$$

We can compare this result with the convergence result of VI. For VI with the supremum norm, following the proof of Equation (2.2) by Munos [2007], we can show that $\|V^* - V^{\pi_k}\|_\infty \leq \frac{2\gamma^k}{1-\gamma}\|V^* - V_0\|_\infty + \frac{2\gamma(1-\gamma^{k-1})\epsilon^{value}}{(1-\gamma)^2}$, with $\|V_i - T^* V_{i-1}\|_\infty \leq \epsilon^{value}$ for all $i < k$ (similar result for the $L_p$-norm also holds, see Theorem 5.2 in Munos 2007). For the approximate VI, the initial error $\|V^* - V_0\|_\infty$ decays with the rate of $O(\gamma^k)$. This should be compared with $O(\gamma'^k)$ rate of OS-VI. The effect of error at each step $\epsilon^{value}$ is also similar: approximate VI has $(1-\gamma)^{-2}$ dependence while approximate OS-VI has $(1-\gamma')^{-2}$. What is remarkable is that as opposed to $\gamma$, which is a fixed parameter of the problem and can be close to 1, $\gamma'$ can be made arbitrary close to zero when the approximate model $\hat{\mathcal{P}}$ becomes more accurate. The additional information given by $\hat{\mathcal{P}}$ allows us to get much faster rate than VI. Of course, this requires the model to be accurate. An inaccurate model might be detrimental to the convergence rate, and may even lead to divergence. Similar conclusions can be made in comparing OS-VI with Policy Iteration and Modified Policy Iteration, as discussed in the supplementary material.

## 5 Operator splitting Dyna

In the RL setting, we only have access to samples from $\mathcal{P}$. MBRL algorithms, such as variants of the Dyna architecture, learn $\hat{\mathcal{P}}$ from those samples, and use it to find the value function or policy. The learned model $\hat{\mathcal{P}}$ is generally different from $\mathcal{P}$ due to the finiteness of the samples as well as the possibility of model approximation error: the true dynamics $\mathcal{P}$ may not be representable with the function approximator used to represent $\hat{\mathcal{P}}$. This is another way to say that the world may be too big to be represented by our models. A MBRL algorithm that uses $\hat{\mathcal{P}}$ in lieu of $\mathcal{P}$ does not find

---

**Algorithm 1** OS-Dyna

---

1: Initialize $V_0, \bar{r} = 0$, and the approximate model $\hat{\mathcal{P}}$.
2: **for** $t = 1, 2, \ldots$ **do**
3:      Sample $(X_t, A_t, R_t, X'_t)$ from environment.
4:      Update the model $\hat{\mathcal{P}}$ with $(X_t, A_t, R_t, X'_t)$.
5:      $\bar{r}(X_t, A_t) \leftarrow \bar{r}(X_t, A_t) + \alpha_t \Big( R_t + \gamma V_{t-1}(X'_t) - \gamma \mathbb{E}_{X' \sim \hat{\mathcal{P}}(\cdot|X_t, A_t)}[V_{t-1}(X')] - \bar{r}(X_t, A_t) \Big).$
6:      $V_t \leftarrow V^{\pi}(\bar{r}, \hat{\mathcal{P}})$ (For PE)    or    $V_t \leftarrow V^*(\bar{r}, \hat{\mathcal{P}})$ , $\pi_t \leftarrow \pi^*(\bar{r}, \hat{\mathcal{P}})$ (For Control).
7: **end for**

---

the true value of the true MDP. Based on OS-VI, we propose OS-Dyna, as a hybrid model-based and model-free RL algorithm, that takes advantage of both the true environment and the model in its updates and can converge to the true value function despite using inaccurate $\hat{\mathcal{P}}$.

Learning $\hat{\mathcal{P}}$ in OS-Dyna is similar to other MBRL algorithms [Moerland et al., 2022]: one can use various model learning approaches, either based on maximum likelihood estimate or a decision-aware model learning approach, to learn the model. Given a learned $\hat{\mathcal{P}}$, we can compute $V_k$ from the auxiliary reward function $\bar{r}_k \triangleq \bar{r}_{V_{k-1}}$ by solving the PE or the Control problem in the auxiliary MDP $(\mathcal{X}, \mathcal{A}, \bar{r}_k, \hat{\mathcal{P}})$, as discussed in Section 3.

As $V_k$ is a function of $\bar{r}_k$, we focus on how $\bar{r}_k$ should be estimated. The update rule of $\bar{r}_k$ in OS-VI entails that for every $(x, a)$, we have

$$\bar{r}_k(x, a) = r(x, a) + \gamma \Big( \mathcal{P}(\cdot|x, a) - \hat{\mathcal{P}}(\cdot|x, a) \Big) V^{\pi}(\bar{r}_{k-1}, \hat{\mathcal{P}}), \qquad \text{(Policy Evaluation)} \quad (5.1)$$

$$\bar{r}_k(x, a) = r(x, a) + \gamma \Big( \mathcal{P}(\cdot|x, a) - \hat{\mathcal{P}}(\cdot|x, a) \Big) V^*(\bar{r}_{k-1}, \hat{\mathcal{P}}). \qquad \text{(Control)} \quad (5.2)$$

We update our estimation of $\bar{r}$ using samples, as shall be discussed soon, and then the value function is updated to $V^{\pi}(\bar{r}, \hat{\mathcal{P}})$ (PE) or $V^*(\bar{r}, \hat{\mathcal{P}})$ (Control) with most recent estimate of $\bar{r}$. The challenge is that the above update rules need access to distribution $\mathcal{P}(\cdot|x, a)$ for every $(x, a)$, while we only have samples from $\mathcal{P}$ at some $(x, a)$ pairs. Fortunately, this challenge has been tackled in developing sample-based algorithms based on the classic VI:

$$\forall (x, a) : \qquad Q_k(x, a) = r(x, a) + \gamma \mathcal{P}(\cdot|x, a) V_{k-1}, \qquad (5.3)$$

where $V_{k-1} = Q_{k-1}(x, \pi(x))$ in PE and $V_{k-1} = \max_{a'} Q_{k-1}(x, a)$ in Control. There are multiple approaches to develop sample-based algorithms based on (5.3) such as Fitted Value Iteration and Stochastic Approximation (SA) [Borkar, 2008]. In this paper we use SA to develop OS-Dyna, but we point out that other algorithms and techniques can also be applied to develop other versions of OS-Dyna. The key step in SA is to use samples to form an unbiased estimate of the intended update value. For a step in the true environment leading to $(X_t, A_t, R_t, X'_t)$ tuple, we can have the estimate $Y_t = R_t + \gamma V(X'_t) - \gamma \mathbb{E}_{X' \sim \hat{\mathcal{P}}(\cdot|X_t, A_t)}[V(X')]$, where the expectation can also be estimated by samples from $\hat{\mathcal{P}}(\cdot|X_t, A_t)$. This estimate $Y_t$ of the update rule can then be used to update $\bar{r}$. As an example, for a finite state-action problem, the update rule is

$$\bar{r}(X_t, A_t) \leftarrow \bar{r}(X_t, A_t) + \alpha_t (Y_t - \bar{r}(X_t, A_t)), \qquad (5.4)$$

where $\alpha_t$ is the learning rate. The final procedure of OS-Dyna is presented in Algorithm 1.

# 6 Experiments

We evaluate both OS-VI and OS-Dyna in a finite MDP and compare them with existing methods. Here we present the results for the Control problem on a modified cliffwalk environment in a $6 \times 6$ grid with 4 actions (UP, DOWN, LEFT, RIGHT). We postpone studying the PE problem, the results for other environments, and other relevant details to the supplementary material. Our convergence analysis shows that the convergence rates of our algorithms depend on the accuracy of $\hat{\mathcal{P}}$. To test OS-VI and OS-Dyna with models of different accuracies, we introduce the smoothed model $\hat{\mathcal{P}}$ of transitions $\mathcal{P}$ with smoothing parameter $\lambda$ as

$$\hat{\mathcal{P}}(\cdot|x, a; \mathcal{P}, \lambda) = (1 - \lambda)\mathcal{P}(\cdot|x, a) + \lambda U\big(\{x'|\mathcal{P}(x'|x, a) > 0\}\big), \qquad (6.1)$$

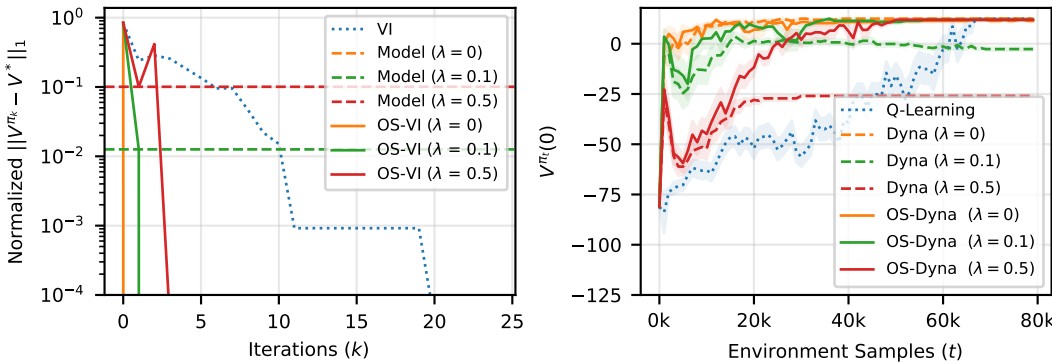

Figure 2: *(Left)* Normalized error comparisons of OS-VI, VI, and the optimal policy of model $\hat{\mathcal{P}}$. *(Right)* Comparison of OS-Dyna with Dyna and Q-Learning in the RL setting.

where $U(A)$ for some set $A$ is the uniform distribution over $A$. Here, $\lambda$ allows making adjustments to the amount of error introduced in $\hat{\mathcal{P}}$ w.r.t. $\mathcal{P}$. If $\lambda = 0$, $\hat{\mathcal{P}} = \mathcal{P}$ will be the accurate model, and if $\lambda = 1$, $\hat{\mathcal{P}}$ will be uniform over possible next states in $\mathcal{P}$.

The left plot in Figure 2 shows the convergence of OS-VI compared to VI and the solutions the model itself would lead to. The plot shows normalized error of $V^{\pi_k}$ w.r.t $V^*$, i.e., $\|V^{\pi_k} - V^*\|_1/\|V^*\|_1$. It can be seen that OS-VI has faster convergence with more accurate models and introduces acceleration compared to VI across different model errors. Note that the convergence of OS-VI has been achieved despite the error in the model. The dashed lines show how a fully model-based algorithm, which only uses $\hat{\mathcal{P}}$, would obtain a suboptimal solution.

We also compare OS-Dyna with Dyna and Q-Learning in the RL setting. At each iteration $t$, the algorithms are given a sample $(X_t, A_t, R_t, X_t')$ where $X_t, A_t$ are selected uniformly at random. For OS-Dyna and Dyna we use the smoothed Maximum-likelihood Estimation (MLE) model. If $\mathcal{P}_{\text{MLE}}$ is the current MLE estimation of the environment transitions, OS-Dyna and Dyna use $\hat{\mathcal{P}}(\mathcal{P}_{\text{MLE}}, \lambda)$ defined in (6.1) as their models. The learning rates are constant $\alpha$ for iterations $t \leq N$ and then decay in the form of $\alpha_t = \alpha/(t - N)$ afterwards. We have fine-tuned the learning rate schedule for *each* algorithm separately for the best results.

The right plot in Figure 2 shows the results for the RL setting. We evaluate the expected return of the policy at iteration $t$ in the initial state of the environment, i.e., $V^{\pi_t}(0)$. Again, OS-Dyna has converged to the optimal policy much faster than Q-Learning. Unlike OS-Dyna, Dyna has failed to find the optimal policy in presence of model error. The results show that OS-Dyna can effectively converge faster than Q-Learning without introducing bias to the final solution due to model error.

## 7 Conclusion

This paper introduced the Operator Splitting Value Iteration (OS-VI) algorithm, which can benefit from an approximate model $\hat{\mathcal{P}} \approx \mathcal{P}$ to accelerate the convergence of the approximate value to the true value function in terms of the number of queries to the true model $\mathcal{P}$. With a small model error, its convergence rate is exponentially faster compared to well-known dynamical programming algorithms such as Value Iteration and Policy Iteration. We also proposed OS-Dyna as a hybrid model-based/model-free algorithm that can bring in the benefits of a model-based RL algorithm without converging to a biased solution, as Dyna or any other purely model-based RL algorithm does. This paper opens up several future directions. Empirically studying the algorithms on problems with large state spaces, for which a function approximator such as a DNN is required, is an obvious one. This is postponed to a future work as our aim was to build the mathematical foundation and conducting experiments without worrying about challenges such as the optimization of a DNN. There are other algorithmic and theoretical directions to be pursued. One is exploring the space of splittings of $\mathbf{I} - \gamma \mathcal{P}^\pi$. The other is whether we can design Operator Splitting variants of other DP algorithms such as Policy Iteration and Modified Policy Iteration, and study their convergence behaviour.

## Acknowledgments and Disclosure of Funding

We would like to thank the anonymous reviewers for their comments that helped us to improve the clarity of the paper, as well as other members of the Adaptive Agents Lab who provided feedback on a draft of this paper. AMF acknowledges the funding from the Canada CIFAR AI Chairs program, as well as the support of the Natural Sciences and Engineering Research Council of Canada (NSERC) through the Discovery Grant program. AR was partially supported by Borealis AI through the Borealis AI Global Fellowship Award. Resources used in preparing this research were provided, in part, by the Province of Ontario, the Government of Canada through CIFAR, and companies sponsoring the Vector Institute.

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
