# OpenReview forum: "Operator Splitting Value Iteration"
_NeurIPS.cc/2022/Conference — NeurIPS 2022 Accept_

### Official Review · Reviewer_nqCV · 2022-06-30

**Rating:** 8
**Confidence:** 4
**Soundness:** 4 excellent
**Presentation:** 4 excellent
**Contribution:** 4 excellent

**Summary:**

This paper considers the problem of planning in a discounted MDP when an approximate model of the dynamics is available. The author propose an algorithm that can take advantage of such model to quickly evaluate a policy or compute an optimal one.

Each iteration of the value iteration algorithm can consists of solving a linear system (in discrete MDPs). In the case where two models are available, the matrix to inverse can be expressed as a difference of two matrices: one involving the approximate transition, one involving the model error. It is shown that solving the original problem is equivalent to applying value iteration in a modified MDP with the approximated dynamics, and a new reward function including the model error.

A theoretical analysis of the convergence of the proposed method is provided along with some extension to a model-based reinforcement learning algorithm.

Experimental results on toy problems are provided and show that the proposed method allow for faster convergence in both planning and RL settings.


**Questions:**

1. How does the state space (discrete vs continuous) affect the method?
2. How do the experimental results look like for more extreme cases when the model is completely off?
3. Would it be possible to get a convergence rate for OS-dyna? In terms of number of sample to reach a certain model accuracy.


**Limitations:**

As far as I understood, this method is only applicable to problem with discrete state spaces, so that the transition model can be represented as a matrix, but it is never clearly stated. The author should make mention of this limitation or clarify how continuous spaces could be handled. The operators are defined for continuous spaces (e.g. 3.3), but I don’t see how the matrix $(I – \gamma P)$ would be represented in such case.

The authors clearly explains that the convergence rate is highly dependent on the quality of the approximate model.


**Strengths And Weaknesses:**

Overall the paper is clearly written and easy to follow.

The idea of considering value iteration through the lens of operator splitting is very interesting and original. The derivation is mathematically correct and very well explained.
The main contribution is theoretical.

Having approximate model of the environment might be common in many applications. The author also show that when no model is given, but the model is learned (OS-dyna) the method still outperform existing model based RL.

The extension to model based RL and the learning of the new reward signal is very clear and a very interesting idea.

It is a bit puzzling that the convergence rate can be made faster, I wish the author added an illustrative example to get a better grasp of this in addition to theorem 2.

minor weaknesses:
- the lack of discussion about continuous state spaces.
- The author could provide a more detailed pseudo code of the method in appendix as algorithm 1 relies on a lot of notation.
- More discussion on how to update the model p_hat would have been useful.

---

> ### Author Response · Authors · 2022-08-02
> **Response (Part 1) - Added visualization + Discussion of Continuous problems**
>
> We thank the reviewer for their valuable feedback and the positive evaluation of the paper. We have added a more detailed algorithm box of OS-Dyna along with a short discussion on model learning to the Appendix C.2.
>
> The addition of an illustrative example is a great suggestion. We have added a visualization in Appendix C.1 that may provide a more intuitive understanding of what OS-VI does. If possible with the space limitations, we plan to move this section to the main text in the camera-ready version. Understanding the dynamics of OS-VI in full detail, beyond the convergence rate that is already provided, is an exciting research question, which we postpone to a future work.
>
>
> **Q: How does the state space (discrete vs continuous) affect the method? As far as I understood, this method is only applicable to problem with discrete state spaces, so that the transition model can be represented as a matrix, but it is never clearly stated. The author should make mention of this limitation or clarify how continuous spaces could be handled. The operators are defined for continuous spaces (e.g. 3.3), but I don’t see how the matrix $(I–\gamma P)$  would be represented in such case.**
>
> A: We would like to clarify which aspect of this paper is specific to discrete state/action spaces and which aspects are applicable to continuous spaces.
> The OS-VI algorithm (Section 3) and its theoretical analysis (Section 4) are for generic state space, discrete or continuous. In fact, theoretical guarantees are specifically for the case of approximate variants of OS-VI (see Equations (4.5) and (4.6) ), where approximation can be due to the use of function approximation or estimation errors (see the first paragraph of Section 4.1). This should be compared with approximate value or policy iteration algorithms.
>
> The OS-Dyna method (Section 5), however, is presented specifically for finite state problems, as the stochastic approximation update (5.4) is not written to consider the function approximation. This could be extended to use a function approximation, but we decided against it to make the presentation of the main idea simple.
> The experiments are all conducted for finite state problems to avoid dealing with complications caused by the use of function approximator such as a deep neural network, which would not add much to the main message and ideas of the paper.
>
> You are right that the definition of the Varga operator has $(I - \gamma \hat{P})^{-1}$ in it, and the computation of $S^\pi V_{k-1}$ (Eq. 3.2) or $S^* V_{k-1}$ (Eq. 3.5) requires the inversion of $(I - \gamma \hat{P}^{\pi})^{-1}$ or $(I - \gamma \hat{P}^{\pi_k})^{-1}$.
> If $\hat{P}$ is a small (stochastic) matrix, we could compute the inversion easily. But for continuous state spaces, $\hat{P}$ would be a transition probability kernel, which is an infinite dimensional operator. In that case, we do not take inverse directly (as it is often infeasible, unless for specific cases), but instead treat it as solving a new MDP that has the dynamics $\hat{P}$ and reward $\bar{r}_k$. This is discussed in the penultimate paragraph of Section 3.1 (give LL numbers).
> If the state space is continuous, solving this new MDP requires approximation. Here, we can use any approximate planning algorithm such as Approximate Value Iteration (or Fitted Q/Value Iteration in the RL setting, including specific algorithms such as DQN or SAC), Approximate Bellman Error Minimization, or whatever other approximate planner that is available to us. Of course, this causes error, whose effect is already incorporated in our error model (see Eq. (4.1) for PE and Eq. (4.5) for Control) and analyzed in our error propagation theorems (Theorems 1 and 2).

---

> > ### Author Response · Authors · 2022-08-02
> > **Response (Part 2) - More extreme experiments + Discussion on the convergence rate for OS-Dyna**
> >
> > **Q: How do the experimental results look like for more extreme cases when the model is completely off?**
> >
> > A: We have added $\lambda = 0.8$ and $\lambda = 1$ to all the plots in the appendix.
> > Please refer to Figures 7 in Appendix E.3 (Effect of model error on OS-VI). As one can see, the convergence rate may be slow or even diverging in extreme cases. But in a large range of parameters, we still have acceleration.
> >
> > **Q: Would it be possible to get a convergence rate for OS-dyna? In terms of number of sample to reach a certain model accuracy.**
> >
> > A: We believe so, but we have postponed such analysis to a future work.
> > The general line of attack is to use tools from statistical learning theory to provide guarantees on the model error and fitting of value function. Those together with our error propagation guarantees lead to the analysis of the OS-Dyna algorithm. This general path is similar to the work done by previous work in the analysis of Fitted Value Iteration, Bellman Residual Minimization, etc., such as [Munos and Szepesvari, 2008], [Antos et al., 2008], [Farahmand et al., 2016], [Chen and Jiang, 2019], as we note in the first paragraph of Section 4.1
> >
> > As a relevant side note, we would like to mention that our convergence rate for the $L_4$ type of guarantee is based on $\chi^2_{\rho}(\mathcal{P}^\pi ||  \hat{\mathcal{P}}^\pi )$ as a measure of model error. An advantage of this form is that it is the expectation of a loss function over a sampling distribution. In this case, the loss is the $\chi^2$-divergence of the model’s distribution and the true distribution. We hope that tools from statistical learning theory help us in bounding this model error.

---

> > > ### Comment · Reviewer_nqCV · 2022-08-03
> > > **Thank you for your reply**
> > >
> > > Thank you for your responses and the clarification on the handling of continuous state spaces. The update to the appendix are appreciated as well.

---

### Official Review · Reviewer_2Dfm · 2022-07-11

**Rating:** 7
**Confidence:** 4
**Soundness:** 3 good
**Presentation:** 3 good
**Contribution:** 3 good

**Summary:**

In this paper, the authors consider a novel Value Iteration by Operator Splitting for both Policy Evaluation and Control problems. Further, they show a faster convergence rate when the learning model is accurate compared with general RL methods with fixed $\gamma$. Then, they give the error bound of the proposed OS-VI and develop OS-Dyna. The experimental environment is simple but effective to show the performance of OS-Dyna.

**Questions:**

Please see the questions in Weaknesses and the following.
	For the proposed algorithm OS-Dyna, the authors claim that the operator splitting approach is a hybrid of MBRL and model-free RL. I agree with this claim. But, I think the authors should add some analysis of the proposed algorithm from the following perspectives.
       a. Compared with the original Dyna framework, they utilize the learning dynamic model $\hat{P}^\pi$ to rollout virtual samplings for improving policy efficiency. How about OS-Dyna’s performance?
       b. Compared with general model-free RL methods, like Soft-Actor-Critic, OS-Dyna utilize the learning dynamic model $\hat{P}^\pi$ as a reward penalty term defined in (3.3). Thus, compared with SAC's entropy policy, how does the proposed penalty term work?
       c. Can OS-Dyna be extended in non-linear systems? Like some general technologies in RL, the authors may consider the sample-based model $\hat{P}^\pi$ and replay buffer to re-sample some experiences for approximately computing the term $P-\hat{P}^\pi$.


**Limitations:**

I think that this work would be more convincing if the authors can extend the performance of OS-VI in some complex environments due to its potential performance improvement.

**Strengths And Weaknesses:**

Strengths
	A novel value iteration formulation to balance Dyna and MFRL.
	Precise problem formulation and derivation structure;
	Reasonable experiments and demonstratable results;
	Clear structure and understandable writing.

Weaknesses
	As mentioned in 2.2, the dynamic error $e_k$ are (norm-) convergent when $\|G\|=\|M^{-1}N\|<1$. When the OS-VI method is proposed, the authors should also discuss the convergence for $M^\pi$ and $N^\pi$ in 3.1, even though they can only be discussed in the linear systems.
	In Theorem 1, the authors give the effective discount as $\gamma’$. I wonder that is there such a situation about $\gamma’>1$. And, if $\gamma’>1$, the proposed Varga operator may not be satisfied (3.2).
	In the proof of Lemma 4, the authors derive $S^\pi V_1- S^\pi V_2$ with some doubts. In different iteration epochs, the Varga operator may correspond to the different learning dynamic model $\hat{P}^\pi$, i.e.,
$S^\pi V_i = (I-\gamma \hat{P}_i^\pi)(r^\pi+\gamma(P^\pi-\hat{P}_i^\pi))$, which may result different result.
	In the experiments, the authors may consider putting the smooth $\lambda$ cases in the paper other than in the supplementary materials.

---

> ### Author Response · Authors · 2022-08-02
> **Response to Reviewer 2Dfm - Strengths And Weaknesses [1/2]**
>
> We would like to thank you for the thorough and detailed feedback, as well as your positive evaluation of this paper. We hope the following answers address your concerns. Please let us know if there is any unclear answer, or any further questions.
>
> **Q: As mentioned in 2.2, the dynamic error $e_k$ are (norm-) convergent when $|G|=|M^{-1}N|<1$. When the OS-VI method is proposed, the authors should also discuss the convergence for $M^\pi$ and $N^\pi$ and in 3.1, even though they can only be discussed in the linear systems.**
>
> A: It is true that the convergence of a splitting procedure depends on the values of $M, N$, and this convergence should be studied for the new choices we are proposing in this paper. We have done this analysis. In fact, this analysis is one of our main contributions. There is no immediate discussion about the convergence in Section 3.1 since the section is meant to just introduce the algorithm. The convergence is discussed in Section 4.
>
>
> We are not sure if we unambiguously understand the comment *“they can only be discussed in the linear systems”*.
>
> If this should be interpreted as we **only** consider the Policy Evaluation problem, where we have to solve a linear system of equations (similar to Section 2.2), we would like to add that this is not the case. We analyze the Control setting too, which can be seen as iterative solving of a nonlinear system of equations. This is done in Section 4.2 and specifically Theorem 2.
>
> If this should be interpreted as we **only** consider MDPs that are described by a linear dynamical system (similar to LQR problem), this is not the case either. We do not make any such assumption in the paper.
>
> We would appreciate it if you clarify this issue, and let us know if we haven’t answered your question.
>
> **Q: In Theorem 1, the authors give the effective discount as $\gamma’$. I wonder that is there such a situation about $\gamma’>1$. And, if $\gamma’>1$, the proposed Varga operator may not be satisfied (3.2).**
>
> A: If the model is inaccurate, it is possible that $\gamma’$ becomes larger than 1. In that case, the Varga operator is still well-defined (notice that it involves the inversion of $(I - \gamma \hat{P})$, which is always invertible), but repeated application of the Varga operator may lead to divergence. In fact, our analysis suggests that divergence might happen in the worst-case (as mentioned in the last sentence of Section 4).
>
> We have observed in practice that even if the theoretical condition for convergence is violated, the algorithm may converge, and in fact, converge with an accelerated rate (see Appendix E.3 for the cases of convergence even in extreme case of model error, as well as some cases of divergence).
> This discrepancy is because the theoretical analysis considers the worst-case scenario, which may not happen in practice. Note that worst-case analysis is not specific to our method and analysis, and is shared by many other similar theoretical results in the literature.
> We would also like to mention that the possibility of divergence when the error is not small is typical in many approximate methods such as Approximate Value Iteration [Munos, 2007], which is the basis of successful algorithms such as DQN.
>
> **Q: In the proof of Lemma 4, the authors derive $S^\pi V_1- S^\pi V_2$ with some doubts. In different iteration epochs, the Varga operator may correspond to the different learning dynamic model ,$\hat{P}^\pi$ i.e., $S^\pi V_i = (I-\gamma \hat{P}_i^\pi)(r^\pi+\gamma(P^\pi-\hat{P}_i^\pi))$, which may result different result.**
>
> A: The theoretical results are for the case that an approximate model is given and fixed throughout iterations. This should address the reviewer's concern about the soundness of the proofs.
>
> The extension to the case that a different model is used in each iteration is straightforward, but complicates the presentation of the analysis. In policy evaluation,
>
> $||V^\pi - V_{k}|| \le  \gamma_k'||V^\pi - V_{k-1}|| + \epsilon $
>
> where $\gamma’_k$ is the effective discount factor for the model in iteration $k$.
> The control case requires more careful treatment, but essentially one needs to add the proper subscript to all $\gamma’$ in the proof.
> The results for the general case of changing models are more complicated. Specifically, a single convergence rate like the current $\gamma’$ will not be apparent in the result. Since the current simplicity of the results allows us to better understand the behaviors of OS-VI, we have decided to keep the results for the simple case.
>
> **Comment: In the experiments, the authors may consider putting the smooth $\lambda$ cases in the paper other than in the supplementary materials.**
>
> A: We appreciate the feedback. We will try to make this change in the camera-ready version with the less strict space limitation.
>
> References:
> Munos, R. (2007). Performance bounds in l_p-norm for approximate value iteration. SIAM journal on control and optimization, 46(2), 541-561.

---

> > ### Author Response · Authors · 2022-08-02
> > **Response to Reviewer 2Dfm - Strengths And Weaknesses [2/2]**
> >
> > **Q: For the proposed algorithm OS-Dyna, the authors claim that the operator splitting approach is a hybrid of MBRL and model-free RL. I agree with this claim. But, I think the authors should add some analysis of the proposed algorithm from the following perspectives**
> >
> > A: We agree with the reviewer that studying OS-Dyna is a very interesting next step. In this paper we are mainly focusing on analysis of OS-VI, which as we have shown, is a rich problem itself. Our results on OS-VI lay a solid theoretical basis for OS-Dyna as an extension of OS-VI to the sample-based regime. We introduced the simplest form of the OS-Dyna algorithm to exhibit the potential of operator splitting in the RL setting, and empirically evaluated it. More detailed analysis of the OS-Dyna, both theoretical and larger scale experiments, is postponed to a future research.
> >
> > **Q: Compared with the original Dyna framework, they utilize the learning dynamic model $\hat{P}^\pi$ to rollout virtual samplings for improving policy efficiency. How about OS-Dyna’s performance?**
> >
> > A: We have provided an empirical comparison of Dyna and OS-Dyna. The convergence rate is competitive between the two, but most importantly Dyna does not converge to the correct solution in presence of model error. This is an important aspect of our work. We allow the agent to use a wrong model without much of a downside. This is a property that does not exist in other MBRL algorithms, as far as we know. A theoretical comparison of OS-Dyna and Dyna is an interesting question for future research.
> >
> > **Q: Compared with general model-free RL methods, like Soft-Actor-Critic, OS-Dyna utilize the learning dynamic model $\hat{P}^\pi$ as a reward penalty term defined in (3.3). Thus, compared with SAC's entropy policy, how does the proposed penalty term work?**
> >
> > A: The new terms in the reward function in SAC and OS-Dyna play very different roles. SAC adds the entropy of action selection to the rewards to motivate policies that pick all viable actions with equal performance. One can think of this term as a regularizer that will make the learned policy different from the optimal policy in exchange to robustness and exploration benefits.
> >
> > On the other hand, the new term in OS-Dyna can be thought of as an error correction term that fixes the difference between the model and the true dynamics. Its purpose is to make the learned policy and values equal to the true ones even though the dynamics we use is different from the environment.
> >
> >
> > **Q: Can OS-Dyna be extended in non-linear systems? Like some general technologies in RL, the authors may consider the sample-based model $\hat{P}^\pi$ and replay buffer to re-sample some experiences for approximately computing the term $P-\hat{P}^\pi$.**
> >
> > A: Similar to our answer to the Weakness above, we are not completely sure about the meaning of linear systems here. Please refer to our answer above.
> >
> > If this should be interpreted as we **only** consider the Policy Evaluation problem, where we have to solve a linear system of equations (similar to Section 2.2), we would like to add that this is not the case. OS-Dyna can be used in the control problem too. The right plot in Figure 1 shows the result for it.
> >
> > If this should be interpreted as we **only** consider MDPs that are described by a linear dynamical system (similar to LQR problem), this is not the case either. We do not make any such assumption in the paper.
> >
> > For the second part, yes indeed. To update rbar, we need estimates of $\mathcal{P}V$ and $\hat{\mathcal{P}}V$ (see equations (5.1) and (5.2)). Estimating these with samples from $\mathcal{P}$, $\hat{\mathcal{P}}$ is possible just like how it is done in DQN and Dyna-style algorithms.

---

> > > ### Comment · Reviewer_2Dfm · 2022-08-08
> > > **Thank you for your response**
> > >
> > > Thanks to the authors for their thorough response as well as their efforts to revise the paper to make it more clear. I have increased my score by 1.

---

### Official Review · Reviewer_rQAP · 2022-07-14

**Rating:** 7
**Confidence:** 3
**Soundness:** 4 excellent
**Presentation:** 3 good
**Contribution:** 4 excellent

**Summary:**

Based on the concept of operator splitting, the authors analyzes convergence properties of value iteration with approximate dynamics models.

**Questions:**

Does the theory apply to continuous state-action cases?

**Limitations:**

Empirical evaluation is only with a simple cliff wall task.


**Strengths And Weaknesses:**

The theory looks sound.

In addition to the analysis, a new algorithm, operator splitting Dyna is proposed.

---

> ### Author Response · Authors · 2022-08-02
> **Response to Reviewer rQAP**
>
> We would like to thank you for your question and your positive evaluation of this paper.
>
> **Q: Does the theory apply to continuous state-action cases?**
>
> A: The OS-VI algorithm (Section 3) and its theoretical analysis (Section 4) are for generic state space, discrete or continuous. In fact, theoretical guarantees are for the case of approximate variants of OS-VI (see Equations (4.5) and (4.6) ), where approximation can be due to the use of function approximation or estimation errors (see the first paragraph of Section 4.1). This should be compared with approximate value or policy iteration algorithms, which are the basis of many modern RL algorithms.
>
> On the other hand, the OS-Dyna method (Section 5) is presented specifically for finite state problems, as the stochastic approximation update (5.4) is not written to consider the function approximation. This could be extended to use a function approximation, but we decided against it to make the presentation of the main idea simple. The experiments are also all conducted for finite state/action problems to avoid dealing with complications caused by the use of function approximators such as a deep neural network, which would not add to the main message of the paper. Extending the OS-Dyna and the experiments to large-scale problems where the use of function approximation is needed shall be a future research direction for us.
>
> A possible source of this confusion might be the fact that matrix splitting, the original inspiration of our algorithm (Section 2.2) is often studied in the context of linear systems of equations in numerical linear algebra, which often deals with finite dimensional systems. We go beyond a finite-dimensional representation by using the operator notation. That is why we have decided to name the algorithm Operator Splitting VI instead of Matrix Splitting VI. We added a note in Section 3.1 after (3.1). The new text added is as follows:
>
> *“Although splitting is originally studied mostly in the context of linear algebra and matrices, we are applying the idea more generally. We are not assuming that the state space $\mathcal{X}$ is finite, and allow it to be more general, such as a subset of $\mathbb{R}^d$. Consequently, $M^\pi$, $N^\pi$, $\mathcal{P}^\pi$, etc. are operators rather than matrices.”*

---

### Author Response · Authors · 2022-08-02
**General Answer to Reviewers and Summary of Revisions**

We would like to thank all the reviewers for their valuable comments and feedback. We are thrilled that reviewers have found our idea novel and interesting (2Dfm, nqCV), the theoretical results sound and well explained (nqCV, 2Dfm, nqCV), experiments effective (2Dfm), and our writing clear (2Dfm).

We answer each of the reviewers separately. We have already incorporated some of their feedback in the revised version of the paper (we use blue color to highlight the changes, except minor ones). We will incorporate the rest in the next iteration of the paper. The significant changes in current revision are:
- New Section C in the appendix to provide visualizations that give an intuitive view of the algorithm as well as a more detailed algorithm box.
- Adding more extreme model errors to all experiments in the appendix.
- New footnotes to clarify the generality of our theoretical results for both discrete and continuous states.

One concern has been brought up by more than one reviewer, so it is worth mentioning it and addressing it here.

**Q: Is the method/theory applicable to continuous state problems?**

A: We would like to clarify which aspects of this paper can deal with continuous state/action spaces, and which aspects are specific to finite state problems.
The OS-VI algorithm (Section 3) and its theoretical analysis (Section 4) are for generic state spaces, discrete or continuous. In fact, theoretical guarantees (Theorem 1 and 2) are for the case of approximate variant of OS-VI (see Equations (4.5) and (4.6) ), where approximation can be due to the use of function approximation and estimation errors (see the first paragraph of Section 4.1). This type of result should be compared with guarantees for approximate value or policy iteration algorithms, which are the basis of modern RL algorithms that can work with high-dimensional state spaces, such as DQN and its variants.

The OS-Dyna method (Section 5), however, is presented specifically for finite state/action problems, as the stochastic approximation update (Eq. 5.4) is not written to consider the function approximation. This could be extended to use a function approximation too, but we decided against it to make the presentation of the main idea as simple as possible.

The experiments are also all conducted for finite state/action problems to avoid dealing with complications caused by the use of function approximators such as deep neural networks. The use of DNN would introduce a plethora of problems such as whether the optimization of the DNN has converged to a good minimum point, the choice of architecture and hyper-parameters to tune, etc., which do not add to the main message of this work.

Admittedly, these are important concerns, as already mentioned in Section 7 (Conclusion).

---

### Meta-Review · Area_Chair_iYcW · 2022-08-26

**Recommendation:** Accept
**Confidence:** Certain

**Metareview:**

All reviewers appreciated the strong theoretical contribution of the paper. The idea was evaluated to be very innovative and offers very good theoratical guarantees. The paper is well written and, while the contribution is mainly theoretical, it also offers a basic evaluation of the presented idea. I follow the reviewers with their recommendation.

**Award:**

No

---

### Decision · Program_Chairs · 2022-09-14

Accept